# Validation of Serum Calprotectin Relative to Other Biomarkers of Infection in Febrile Infants Presenting to the Emergency Department

**DOI:** 10.3390/antibiotics13050425

**Published:** 2024-05-07

**Authors:** Mary Kathryn Bohn, Aleksandra Havelka, Mats Eriksson, Khosrow Adeli

**Affiliations:** 1Department of Pathology and Laboratory Medicine, University of Toronto, Toronto, ON M5G 1X8, Canada; khosrow.adeli@sickkids.ca; 2Molecular Medicine, SickKids Research Institute, Toronto, ON M5G 0A4, Canada; 3Department of Molecular Medicine and Surgery, Karolinska Institute, 171 76 Stockholm, Sweden; aleksandra.havelka@gentian.no; 4Gentian AS, 1596 Moss, Norway; 5Department of Surgical Sciences, Section of Anaesthesiology and Intensive Care Medicine, Uppsala University, 751 85 Uppsala, Sweden; mats.b.eriksson@uu.se; 6NOVA Medical School, New University of Lisbon, 1099-085 Lisbon, Portugal

**Keywords:** antibiotic, biomarker, calprotectin, emergency department, infection, neonate

## Abstract

Antimicrobial stewardship involves a delicate balance between the risk of undertreating individuals and the potential societal burden of overprescribing antimicrobials. This balance is especially crucial in neonatal care. In this observational study, the usefulness of biomarkers of infectious diseases (calprotectin, procalcitonin (PCT), C-reactive protein (CRP), and white blood cells (WBCs) were evaluated in 141 febrile infants aged 28–90 days presenting to an emergency department. Since our focus was on the usefulness of serum calprotectin, this biomarker was not part of clinical decision-making. A significant difference was observed in the levels of all biomarkers, related to final discharge diagnosis and disposition status. The difference in levels related to antibiotic prescription was significant for all biomarkers but WBCs. The performance of calprotectin in the detection of bacterial infections (AUC (95% CI): 0.804 (0.691, 0.916)) was comparable to the performance of both PCT (0.901 (0.823, 0.980)) and CRP (0.859 (0.764, 0.953)) and superior to the WBC count (0.684 (0.544, 0.823)). Procalcitonin and CRP demonstrated a statistically significantly higher specificity relative to calprotectin. In this cohort, antibiotic use did not always correlate to a definite diagnosis of confirmed bacterial infection. The sample size was limited due to associated challenges with recruiting febrile infants. Hence, there is a need for adequate diagnostic tools to help discriminate between various kinds of infections. This study suggests serum calprotectin, procalcitonin, and CRP may serve as valuable biomarkers to differentiate between types of infection, in addition to clinical input and decision-making.

## 1. Introduction

Small children who present to an emergency department represent a broad variety of conditions [1,2], from rhinitis to severe infections, including diseases with a high risk of mortality (e.g., epiglottitis, meningitis, or sepsis), where adequate treatment should be initiated as soon as possible. Antibiotic treatment has no therapeutic value in viral disease but may increase the risk of the induction of bacterial resistance, which is a constantly increasing global health issue [3]. Furthermore, children with viral infections who are subjected to unnecessary antibiotic treatment face additional hazards due to antibiotic toxicity or sensibilization.

There is no clear guidance for screening febrile infants to assess the source of infection (e.g., viral or bacterial), although several possibilities exist, each one with its own limitations [4]: The culturing of microorganisms in samples has several advantages but takes several days to complete. PCR provides microbial identification within hours but is fairly expensive and requires high-tech instrumentation. The light microscopy of bodily fluids may provide rapid identification, but its value for screening is limited. Radiological imaging should only be used in selected cases. Biomarkers are rapid and cost-effective for the diagnosis of infectious disease, although specificity is limited.

While avoiding excessive antibiotic exposure is crucial in several aspects, the identification of life-threatening disease is of major importance in the treatment hierarchy. Neonatal sepsis is a major cause of mortality in pediatrics [5]. If neonatal sepsis is left untreated, it can rapidly be fatal. Hence, it is important to identify children at risk at an early timepoint, so that therapeutical interventions may prevent disease progression [6].

In a clinical study comprising 41 infants with suspected sepsis, serum calprotectin turned out to be both a sensitive and a specific marker for sepsis, as evaluated by a positive blood culture [7]. Similar findings were noted in newborns with a very low birth weight, where calprotectin was significantly higher in newborns with confirmed sepsis than in noninfected subjects or healthy controls [8]. Furthermore, serum calprotectin is a reliable biomarker that independently predicts the risk of sepsis in preterm neonates, thereby reducing mortality rates and facilitating antibiotic management [9]. However, there are minimal data investigating its utility in the diagnosis of less severe bacterial infections, including urinary tract infections and bacteremia. 

Calprotectin, a mediator of the innate immune response against infections, is a complex of two proteins, S100-A8 and S100-A9, with local bacteriostatic and cytokine-like effects [10]. Calprotectin is expressed from neutrophil granulocytes when they are activated, and high concentrations of this protein in the blood indicate a severe infection [11]. In severe infections, neutrophil granulocytes adhere to the vessel walls and penetrate them in order to reach an infectious focus. Such a migration frequently lowers the number of neutrophil granulocytes in the blood, which limits both the diagnostic and the predictive value of determining their count in the blood during the acute phase of a severe infection [12].

The objective of this study was to evaluate the clinical performance of serum calprotectin in identifying bacterial infection in febrile infants presenting to an emergency department (ED) relative to other common biomarkers of infection (e.g., C-reactive protein, procalcitonin, and WBC count). 

## 2. Results

### 2.1. Clinical Cohort

The clinical characteristics of the patient population (35% male, 65% female) are reported in Table 1. The chief complaint at ED presentation was fever, with over 90% of patients presenting as well-appearing. Additional complaints included vomiting and/or diarrhea, rash, and shortness of breath. A quantitative summary of the available laboratory test results is presented in Table 1, including urinalysis, blood culture, urine culture, and nasopharyngeal swab results. 

The final diagnosis was considered as bacterial infection (n = 37, including 4 infants with sepsis), viral infection (n = 24), a fever from unknown source (n = 45), and other diagnoses (n = 35, including poor feeding of the newborn, jaundice, seizure, and rash). In one case, a final diagnosis was not noted. Bacterial infection included urinary tract infection (UTI) (n = 27), bacteremia (N = 1), UTI in addition to bacteremia (n = 5), and sepsis (n = 4) per definitions provided in the Materials and Methods section. Of the four patients with sepsis, two were classified as culture-negative sepsis cases. 

Antibiotics were prescribed in 36 cases, including all septic patients, 22 of 33 patients with defined UTI and/or bacteremia, 4 of 45 patients with fever from an unknown source, and 6 of 35 patients with other diagnoses. Antibiotics prescribed included ampicillin, ceftriaxone, tobramycin, amoxicillin, cephalexin, and trimethoprim. Statistically significant differences were observed in serum calprotectin (*p* < 0.001), procalcitonin (*p* < 0.001), and CRP (*p* < 0.001) concentrations during the discharge diagnosis (Table 2). Marked elevations of both calprotectin and CRP were observed, particularly in patients with bacterial infection (Figure 1). 

At ED discharge, 75 (53%) of patients were discharged from The Hospital for Sick Children, while 60 (43%) were hospitalized or transferred to another hospital. Statistically significant increases in the serum concentrations of all biomarkers were observed in hospitalized patients relative to those discharged: calprotectin (*p* < 0.05), procalcitonin (*p* < 0.001), CRP (*p* < 0.001), and WBCs (*p* < 0.05) (Table 2, Figure 2). This should be interpreted with caution, as the levels of these biomarkers, except calprotectin, likely guided the decision to admit. Significant increases in calprotectin, procalcitonin, and CRP were also observed in patients prescribed antibiotics (74%), relative to those without antibiotic treatment (Table 2, Figure 3). No differences in biomarker concentrations were observed in term-relative to pre-term patients. In six cases, the follow-up on biomarkers was lost.

### 2.2. Biomarker Performance

A ROC curve analysis was completed for the identification of bacterial infection (including sepsis cases) in the entire clinical cohort (Table 3, Figure 4A, n = 141) and in only patients with laboratory-confirmed bacterial or viral infection (Table 4, Figure 4B, n = 57) through urine and blood cultures. Calprotectin demonstrated the highest sensitivity (78.3%) relative to procalcitonin (67.6%), CRP (64.9%) and WBC count (62.1%) in the entire cohort; however, this was not statistically significant. When considering only patients with confirmed bacterial or viral infection, sensitivities and specificities differed across biomarkers. The AUC for calprotectin for the entire cohort was 0.767 and, for patients with confirmed infections, 0.804. The AUC for procalcitonin for the entire cohort was 0.790 and, for patients with confirmed infections, 0.901. No statistically significant differences in ROC curve performance were observed between calprotectin and procalcitonin or CRP per De Long’s test (*p* > 0.05) (Table 3 and Table 4, Figure 4). Procalcitonin and CRP demonstrated a statistically significantly higher specificity relative to calprotectin. Youden’s index for all biomarkers was only marginally above 0.5, with the exception of procalcitonin. Of note, all the biomarkers except calprotectin were used in the routine clinical adjudication, and thus a direct head-to-head comparison may be biased in favor of these biomarkers. 

## 3. Discussion

This observational study was performed on a cohort of 141 infants aged 28–90 days with suspected infectious disease, where 36 were prescribed antibiotic therapy at the time of initial examination. The clinical picture varied considerably among these children, but it must be emphasized that, in neonates, sepsis is a life-threatening infection. Bacterial infections were only confirmed in 37 cases and did not always correlate with antibiotic use. Unnecessary antibiotic use should, by definition, be avoided, but neglected antibiotic prescribing may also have negative consequences [13].

The decision to prescribe antibiotics for infants is challenging [14]. In a clinical survey across 11 countries, antibiotics were prescribed in a ratio between 19% and 64% for febrile children aged between 1 month and 16 years [15]. In febrile infants, the biomarkers of infection (i.e., urinalysis, absolute neutrophil count, and procalcitonin) have the possibility of acting as an adjunct in reducing unnecessary and potentially harmful diagnostic procedures, antibiotic administration, and hospitalizations [16]. 

Few studies have evaluated the utility of serum calprotectin, relative to other biomarkers of infection, in infants presenting to the emergency department. A similar study, performed in a cohort of adult intensive care (ICU) patients, revealed that calprotectin was an early and specific marker of bacterial infection, with a better predictive value than WBCs and PCT [17]. The cut-off established in this study, of 1.80 mg/L, is similar to that established in our investigations. Ideally, a biomarker intended to differentiate between bacterial and viral infections should have both a 100% sensitivity and specificity. However, biological processes are complex, and the prospect of a single test being able to make such a differentiation, particularly when considering the time of symptom onset, seems very distant. The optimization of the cut-off values for particular biomarkers may allow for a prioritization of sensitivity to avoid missing serious bacterial infection or the specificity to avoid unnecessary antibiotic prescription. The role of neutrophilic granulocytes in bacterial disease is well-established, but granulocytes are also activated in viral infections [18]. Hence, calprotectin is also expressed in viral infections, although the levels of this protein are, in general, significantly higher in bacterial infections [19,20]. This is highlighted by the moderate performance observed for all biomarkers in our cohort in the differentiation of bacterial and viral infection. Moreover, calprotectin may be used to assess disease severity, being more elevated in patients with severe disease requiring hospital admission and/or transfer to a higher level of care. A study, performed in adult patients presenting to the ED and with suspected sepsis demonstrated that calprotectin was superior to CRP, procalcitonin and neutrophil/lymphocyte ratio in predicting the need for transfer to the ICU and other higher level of care [21]. In our study, calprotectin showed a good performance in the detection of bacterial infections and the differentiation between bacterial and viral infections, with an AUC of 0.804. However, we did not assess the effect of disease severity on biomarker levels, due to a low sample size. A similar performance of calprotectin in the detection of bacterial infections has been reported in other studies, confirming the potential value of the biomarker in adults and children [17,20]. Calprotectin, available as a turbidimetric assay on high-throughput clinical chemistry analyzers, may therefore provide fast and accurate results and reduce the time taken to make decisions on clinical management [22,23]. 

New biomarkers are being increasingly incorporated into guidelines for the evaluation of fever with an unknown source in neonates. In particular, the American Academy of Pediatrics (AAP) clinical practice guidelines published in 2021 incorporated procalcitonin into the algorithm for the identification of bacterial infection in well-appearing febrile infants [24]. This was based on several lines of evidence supporting value in this population. For example, a prospective study of 1821 febrile infants with 30 cases of invasive bacterial infections found an elevated neutrophil count, combined with abnormal urinalysis and a procalcitonin of greater than 1.7 ng/mL, detected 29 of 30 cases (sensitivity: 96.7%; sensitivity: 61.5%) and no missed cases of meningitis [16]. Similarly, in a prospective study of several EDs, a procalcitonin cut-off of 0.3 ng/mL was shown to assist in the stratification of low- and high-risk infants for invasive bacterial infection, with an AUC of 0.91 [25]. These findings are in line with our results. However, it is important to note that the cut-off established in our study (0.09 ng/mL), which identified all cases of sepsis and UTI plus bacteremia, is much lower relative to other literature and AAP guidelines (0.50 ng/mL). According to international guidelines, pediatric sepsis refers to an infection with life-threatening organ dysfunction [26,27,28]. Thus, a positive blood culture in the absence of such an organ dysfunction was not considered to be equivalent to “sepsis”. This highlights that lower concentrations may also be associated with serious bacterial infection. When comparing calprotectin to procalcitonin, no statistically significant differences were observed between the AUC estimates (*p*-value = 0.707); however, the AUC of calprotectin was slightly lower. 

### Strengths and Limitations

This infant cohort, uniform in age, represents a broad spectrum of various diagnoses, suggesting that our results may be relevant from a broad pediatric perspective. The main problem in this context is that a gold standard for differentiating between various infectious diseases is lacking. Expert diagnosis is frequently denoted as a reference standard, although interobserver agreements are imperfect [29]. However, this is a limitation shared between all studies, unless patients are strictly selected for specific diagnostic purposes, e.g., PCR or blood culture [30]. Another drawback is that this is a single-center study, no longitudinal follow-up was performed, and the severity of infection could not be assessed. Comparing biomarker concentrations at admission may not capture the full utility of biomarkers such as calprotectin and procalcitonin in longitudinal patient monitoring and prognostication. An assessment of the kinetics of these biomarkers was outside the scope of this study, particularly given concerns of repeat phlebotomy in these patients and the need for rapid differentiation of viral and bacterial infection. 

The sample size was limited, due to the challenges of recruiting young patients aged <90 days; however, it is comparable to other investigations of infectious disease biomarkers in this population and in keeping with the clinical phase II trial definition of 20 to 80 patients [31]. For example, Ramineni et al. included 83 children in their study evaluating biomarkers of neonatal sepsis [32]; Xie et al. succeeded in recruiting 195 children in their study evaluating the diagnosis and severity of community-acquired pneumonia in children [33]. The sample size across other studies ranged from 40 to 116 [34,35]. A limited sample size may influence study conclusions due to type I and type II errors; however, the statistically significant differences observed in our study suggest it was well-powered to assess biomarker performance, particularly in such a challenging population. Additional studies may be needed to confirm the findings in larger cohorts, to strengthen the conclusions and their application to clinical care. 

It is also important to note that this study was conducted during the course of the COVID-19 pandemic. While only two patients had documented COVID-19 infection, the clinical evaluation of viral and bacterial infection in these patients was influenced by the viral species circulating at that time, as well as the enhanced infection prevention precautions in place. Calprotectin has also been suggested to increase in severe COVID-19 infection, due to the activation of TLR4 and higher neutrophil engagement compared to other viral infections [36,37,38]. 

## 4. Materials and Methods

### 4.1. Study Population

One hundred and forty-one infants aged 28–90 days with suspected infectious disease presenting to the ED at The Hospital for Sick Children in Toronto, Canada, were included in this non-interventional investigation. Patient characteristics are seen in Table 1. The study was performed in accordance with the ethical principles of the Declaration of Helsinki [39] and applicable local regulatory requirements. The study was approved by the Research Ethics Board at The Hospital for Sick Children in Toronto, Canada (#1000075791). Since this was an observational and descriptive study, informed consent was not required per our ethics proposal. Medical examinations were performed by qualified pediatricians at the initial ED visit. All biomarkers except the investigated biomarker (i.e., calprotectin) were used in the routine clinical adjudication of the final diagnosis per the following pathway. 

In all well-appearing patients, two clinical decision pathways were followed, depending on patient age [40]. In patients aged 1–28 days, a full diagnostic work-up was completed, including a complete blood count (CBC), urinalysis, urine culture, and serum procalcitonin, as well as CSF cell count, glucose, protein, and culture. If significant respiratory symptoms were observed, a chest X-ray and nasopharyngeal swab were considered. If the investigations were consistent with meningitis, patients were treated with IV ampicillin and cefotaxime, or IV acyclovir if indicated. If the investigations were not consistent with meningitis, bacteremia and/or UTI were ruled out or treated with IV penicillin and tobramycin. In those aged 29–90 days, patients were reviewed for symptoms of bronchiolitis. If there were no symptoms, a full diagnostic work-up was completed, including a complete blood count (CBC), urinalysis, urine culture, serum procalcitonin, CSF cell count, glucose, protein, and culture. If significant respiratory symptoms were observed, a chest X-ray and nasopharyngeal swap were considered. If the urinalysis was positive, patients were diagnosed and treated for UTI. If the urinalysis was not positive, patients with elevations of CRP and PCT underwent a lumbar puncture to evaluate for meningitis. If no elevations were observed, patients were either discharged home, with a 12–24 h follow-up secured or admitted for [40]. Serum calprotectin was not available to clinicians to support clinician decision-making.

A final evaluation of the diagnosis was made either at discharge from the ED or upon their return 72 h later [40]. Four main final diagnoses were considered in the analysis, including bacterial infection, viral infection, fever from an unknown source, and other diagnoses (including poor feeding of the newborn, jaundice, seizure, and rash). In one case, a final diagnosis was not noted. Bacterial infection included urinary tract infection (UTI), bacteremia, UTI in addition to bacteremia, and sepsis. UTI was defined as the presence of leukocytes| ≥ |small/trace|and/or any nitrites on the urinalysis, eventually confirmed with >10| × |10^6^ colony-forming units per liter (CFU/L) of a single pathogen bacteria from a catheterized urine specimen or positive urine culture alone [41]. Bacteremia was defined as a positive blood culture with a pathologic organism (excluding coagulase-negative Staphylococcus). “Sepsis” was diagnosed by the evaluation of clinical symptoms and biomarkers per routine clinical procedure and in agreement with international conventions [27,28]. At discharge, patients with a final diagnosis of fever from an unknown source or other had no evidence of bacterial or viral infection in the urinalysis and/or urine, blood, and/or CSF culture.

Residual serum specimens collected as part of the standard care pathway were stored at −80 °C, prior to the analysis of serum calprotectin using the GCAL^®^ assay. A chart review was completed for key variables, including age (days), gestational age at birth, sex, chief complaint and temperature at ED presentation, and available laboratory test results (i.e., C-reactive protein and procalcitonin (Architect, Abbott Diagnostics, Wiesbaden, Germany), WBC count and neutrophil count (XN3000, Sysmex Europe GmbH, Norderstedt, Germany), urinary leukocytes and nitrates (urinalysis reflectance spectrophotometry), and blood and/or urine bacterial culture). The final diagnosis, admission status, and antibiotic prescription at ED discharge were also extracted, including a return to the ED within 72 h. Clinical care was at the discretion of the treating clinician.

### 4.2. Analytical Validation of Immunoturbidimetric Serum Calprotectin Assay (GCAL)

A three-part analytical validation of the serum calprotectin assay (GCAL^®^, Gentian AS, Moss, Norway) on an Abbott Architect instrument (The Hospital for Sick Children, Toronto, ON, Canada) was completed in accordance with the Clinical and Laboratory Standards Institute (CLSI) EP05-A3, EP06-A, and EP09-A3 guidelines. The assay precision was evaluated using two internal quality control levels run five times per day over a five-day period. Linearity was evaluated by assaying 10 levels with defined concentrations in triplicate across the analytical measuring range (0 to 19 mg/L). Linear regression was used to determine slope, intercept, and R2 agreement. A method comparison was also completed, using 40 patient specimens evaluated by the GCAL^®^ assay at two separate laboratories using Abbott Architect and Roche Cobas c501 instrumentation (Basel, Switzerland). A Deming regression was used to determine the line of best fit, and Pearson’s correlation coefficient was calculated to determine the strength of agreement. Assay analytical characteristics are provided in Appendix A. 

### 4.3. Statistical Analysis

A statistical analysis was performed using R Statistical Software (Version 4.2.3). Significant differences in serum calprotectin, procalcitonin, C-reactive protein, and WBC count between final discharge diagnosis classifications, disposition status, and the use of antibiotic prescription was evaluated visually, using boxplots, and statistically, using the nonparametric Kruskal–Wallis test. A receiver operating characteristic curve (ROC) curve analysis was also completed using Youden’s index and the area under the curve (ROCit R Package). Informedness was calculated as sensitivity + specificity − 1. The AUC for each ROC curve was compared using DeLong’s test for two correlated ROC curves. A *p*-value of <0.05 was considered statistically significant. The sensitivity and specificity across assays were directly compared relative to calprotectin using McNemar’s hypothesis test. This test for paired binary observations is often used to compare the sensitivities and specificities for the evaluation of two diagnostic tests, where the reference test and the new test can be compared by checking the equality of marginal positives, as described elsewhere [42].

## 5. Conclusions

There is a need for the biomarker-based differentiation of viral and bacterial infection. Untreated infections in infants may have devastating consequences. In this study, calprotectin, procalcitonin, and CRP demonstrated a similar performance in the detection of bacterial infections and in distinguishing between bacterial and viral infections in infants aged 1–3 months. Procalcitonin and CRP are recommended in the guidelines for the management of febrile illness, due to several lines of supporting evidence as reviewed above. Calprotectin is a newer biomarker, whose performance showed no statistically significant differences to CRP or procalcitonin in our study. The sensitivity was higher relative to other biomarkers; however, this did not reach statistical significance, likely due to the low sample size. This highlights the complexities of the biochemical investigation of bacterial infections in a representative cohort of infants with fever. Various diagnostic tools may be seen as a palette of possibilities, each one with its own advantages and limitations. Hence, it is not surprising that combining biomarkers with clinical scores seems to be a promising aid in the diagnosis of severe infections, with potential further improvements with the application of machine learning and/or artificial intelligence [43,44].

## Figures and Tables

**Figure 1 antibiotics-13-00425-f001:**
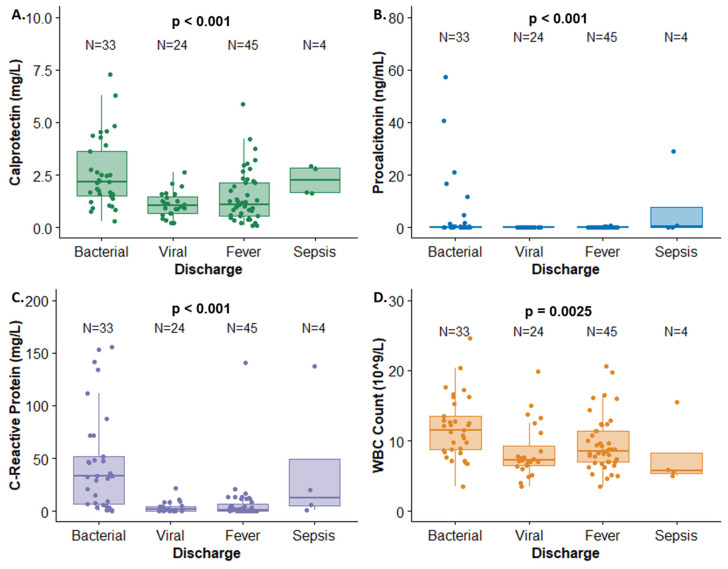
Biomarker concentrations by discharge diagnosis for (**A**) calprotectin, (**B**) procalcitonin, (**C**) C-reactive protein, and (**D**) WBC count. Cases with a final diagnosis classified as other were excluded from this figure.

**Figure 2 antibiotics-13-00425-f002:**
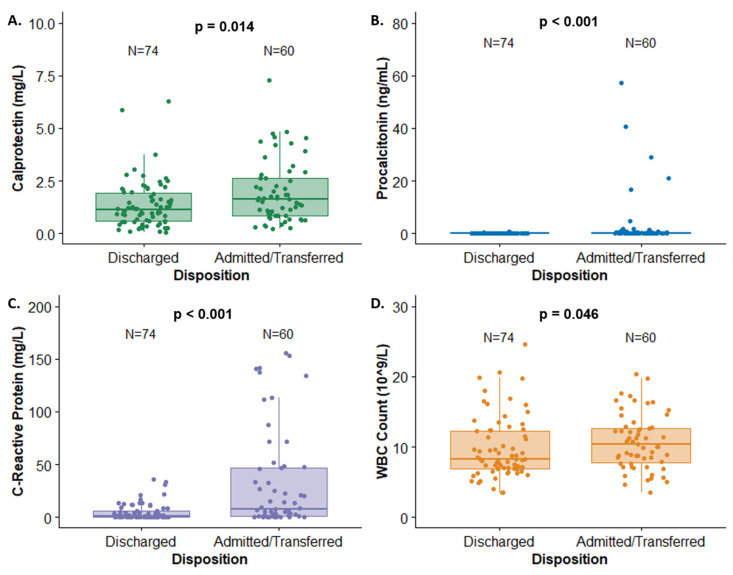
Biomarker concentrations by admission status for (**A**) calprotectin, (**B**) procalcitonin, (**C**) C-reactive protein, and (**D**) WBC count.

**Figure 3 antibiotics-13-00425-f003:**
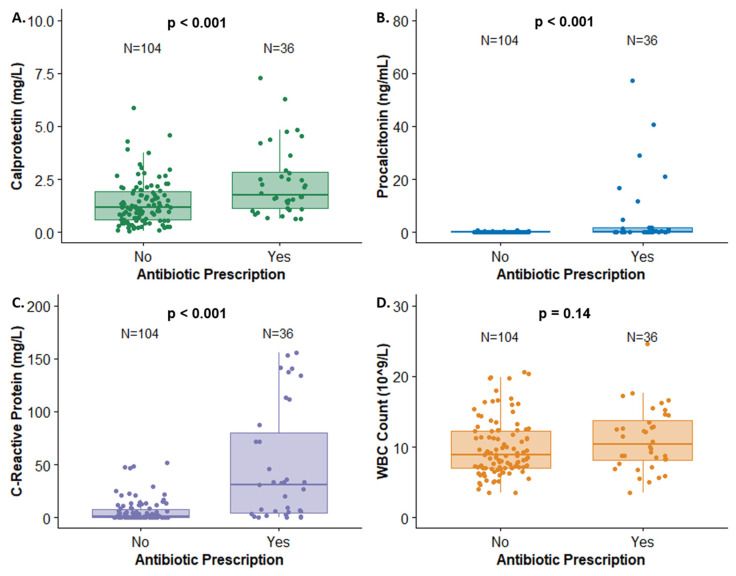
Biomarker concentrations by antibiotic prescription status for (**A**) calprotectin, (**B**) procalcitonin, (**C**) C-reactive protein, and (**D**) WBC count.

**Figure 4 antibiotics-13-00425-f004:**
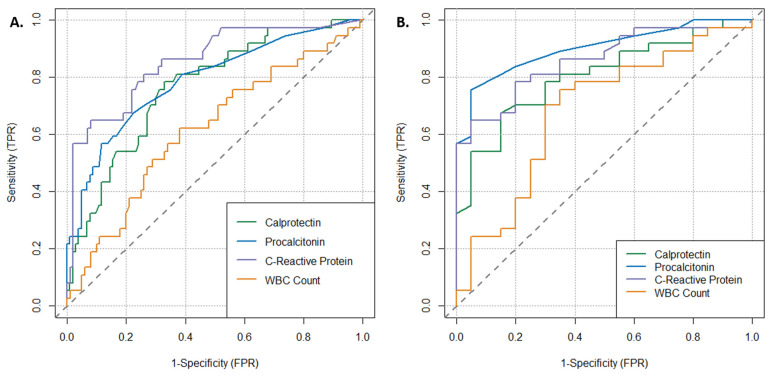
ROC curve analysis for evaluated inflammatory parameters in detection of bacterial infection for (**A**) entire clinical cohort (N = 141) and (**B**) only those with diagnosed and confirmed bacterial or viral infection (N = 41).

**Table 1 antibiotics-13-00425-t001:** Clinical characteristics of patient population at presentation to ED (n = 141).

Patient Characteristics	Summary [N (%) or Median (IQR)]
**Clinical Characteristics**	
**Males**	49 (35%)
**Females**	91 (65%)
**Unknown Sex**	1 (0.7%)
**Gestational Age at Birth (weeks)**	38 (37–38)
Preterm (<37 weeks)	14 (10%)
Term (37–41 weeks)	126 (89%)
Unknown	1 (1%)
**Age at Sample Collection (days)**	51 (34–70)
**Method of Temperature**	
Rectal	131 (91%)
Axillary	4 (3%)
Auricular	1 (1%)
Unknown	5 (4%)
**Chief Complaint at Presentation**	
Fever	108 (77%)
Vomiting and/or Diarrhea	8 (6%)
Rash	3 (2%)
Shortness of Breath	5 (4%)
Other	17 (12%)
**Ill Appearing**	
No	129 (91%)
Yes	11 (8%)
Unknown	1 (1%)
**Laboratory Results**	
**WBC Count (10^9^/L)**	9.2 (7.1–12.6)
**Neutrophil Count (10^9^/L)**	2.7 (1.4–4.6)
**C-Reactive Protein (mg/L)**	2.6 (0.5–13.9)
**Procalcitonin (ng/mL)**	0.07 (0.05–0.15)
**Calprotectin (mg/L)**	1.30 (0.75–2.14)
**Leukocytes (urinalysis)**	
Negative	105 (74%)
Small/Trace	13 (9%)
Moderate	5 (4%)
Large	11 (8%)
Not completed	7 (5%)
**Nitrates (urinalysis)**	
Negative	129 (89%)
Positive	7 (5%)
Not Completed	8 (6%)
**Nasopharyngeal Swab**	
Negative	86 (61%)
Positive	22 (16%)
Not Completed	33 (23%)
**Blood Culture**	
Negative	128 (91%)
Positive	7 (5%)
[*Escherichia coli*]	2
[*Coagulase-negative Staphylococcus*]	2
[*Moraxella catarrhalis*]	1
[*Streptococcus agalactiae*]	1
[*Gram-negative Bacilli*]	1
Not Completed	6 (4%)
**Urine Culture**	
Negative	93 (66%)
Positive	34 (24%)
[*Escherichia coli*]	22 (24%)
[*Enterobacter cloacae*]	3
[*Enterococcus faecalis*]	2
[*Klebsiella pneumoniae*]	2
[*Staphylococcus aureus*]	1
[*Streptococcus agalactiae*]	1
[*Proteus mirabilis and Enterococcus faecalis*]	1
[*Citrobacter koseri*]	1
[*Mixed growth indicative of contamination*]	1
Not Completed	12 (10%)
**CSF Culture**	
Negative	23 (16%)
Not Completed	118 (84%)

**Table 2 antibiotics-13-00425-t002:** Final diagnostic and treatment information of patient population at ED discharge. Biomarker concentrations displayed as median (interquartile range).

Patient Characteristic	N	Calprotectin (mg/L)	Procalcitonin (ng/mL)	C-Reactive Protein (mg/L)	White Blood Cell Count (10^9^/L)
**Final Discharge Diagnosis**					
Bacterial Infection	33	2.16 (1.50, 3.62)	0.17 (0.08, 0.57)	33 (7, 52)	11.5 (8.7, 13.5)
Viral Infection	24	1.05 (0.69, 1.47)	0.05 (0.04, 0.07)	2 (1, 5)	7.2 (6.5, 9.2)
Fever	45	1.08 (0.56, 2.14)	0.07 (0.05, 0.11)	1 (0, 7)	8.5 (7.0, 11.4)
Sepsis	4	2.25 (1.67, 2.84)	0.42 (0.17, 7.76)	13 (5, 49)	5.7 (5.4, 8.3)
Other	35	1.17 (0.62, 1.76)	0.06 (0.04, 0.11)	0 (0, 6)	10.1 (8.3, 13.5)
***p-*****value** **^a^**	-	**<0.001**	**<0.001**	**<0.001**	**0.0025**
**Disposition Status**					
Discharged	75	1.13 (0.61, 1.94)	0.06 (0.05, 0.09)	1 (0, 6)	8.2 (6.9, 12.3)
Admitted/Transferred	60	1.65 (0.86, 2.62)	0.13 (0.06, 0.49)	8 (1, 46)	10.4 (7.8, 12.7)
Unknown	6	1.58 (1.41, 2.39)	0.05 (0.04, 0.14)	3 (0, 24)	11.2 (9.4, 14.0)
***p-*****value** **^a^**	-	**0.014**	**<0.001**	**<0.001**	**0.046**
**Antibiotic Prescription**					
Yes	36	1.77 (1.13, 2.84)	0.18 (0.08, 1.63)	31 (4, 80)	10.4 (8.1, 13.7)
No	104	1.18 (0.61, 1.91)	0.07 (0.05, 0.10)	1 (0, 8)	8.9 (7.1, 12.3)
Unknown	1				
***p-*****value** **^a^**	-	**<0.001**	**<0.001**	**<0.001**	0.14

^a^ *p*-value calculated by nonparametric Wilcoxon sum test (2 subgroups) or Kruskal–Wallis test (3+ subgroups). Unknown/other variables were excluded from comparative analysis.

**Table 3 antibiotics-13-00425-t003:** ROC curve analysis for detection of bacterial infection in entire clinical cohort (N = 141).

	Calprotectin	Procalcitonin	C-Reactive Protein	WBC Count
AUC (95% CI)	0.767 (0.670, 0.864)	0.790 (0.697, 0.790)	0.855 (0.775, 0.937)	0.612 (0.502, 0.721)
Youden Index	0.457	0.454	0.569	0.235
Cut-off	1.50 mg/L	0.11 ng/mL	15.3 mg/L	9.74 × 10^9^/L
Sensitivity (*p*-value ^a^)	78.3%	67.6% (0.248)	64.9% (0.166)	62.1% (0.133)
Specificity (*p*-value ^a^)	67.3%	77.9% (0.048)	92.1% (<0.001)	61.4% (0.330)
Informedness	45.6%	45.5%	57.0%	23.5%

^a^ *p*-value calculated relative to calprotectin performance, using McNemar test for comparison of sensitivities and specificities in paired study design.

**Table 4 antibiotics-13-00425-t004:** ROC curve analysis for detection of bacterial infection vs. viral infection in clinical cohort wherein final diagnosis was confirmed (N = 57).

	Calprotectin	Procalcitonin	C-Reactive Protein	WBC Count
AUC (95% CI)	0.804 (0.691, 0.916)	0.901 (0.823, 0.980)	0.859 (0.764, 0.953)	0.684 (0.544, 0.823)
Youden Index	0.526	0.707	0.598	0.407
Cut-off	1.66 mg/L	0.09 ng/mL	15.3 mg/L	8.23 × 10^9^/L
Sensitivity (*p*-value ^a^)	67.5%	75.7% (0.405)	64.9% (0.781)	75.7% (0.365)
Specificity (*p*-value ^a^)	85.0%	95.0% (0.317)	95.0% (0.317)	65.0% (0.102)
Informedness	52.5%	70.7%	59.9%	40.7%

^a^ *p*-value calculated relative to calprotectin performance, using McNemar test for comparison of sensitivities and specificities in paired study design.

## Data Availability

The datasets presented in this article are not readily available because of ethical approval restrictions.

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
