# Peer review of "Validation of Serum Calprotectin Relative to Other Biomarkers of Infection in Febrile Infants Presenting to the Emergency Department"

_antibiotics, 2024, doi:10.3390/antibiotics13050425_

Round 1
Reviewer 1 Report
Comments and Suggestions for Authors
Dear Authors,
I hope this message finds you well.
I am writing to share my feedback on your manuscript titled "Clinical utility of serum biomarkers of infection in febrile infants presenting to the Emergency Department." After careful consideration, I regret to inform you that I do not believe the article is suitable for publication in a journal of high standards such as Antibiotics.
Upon review, I found that the manuscript lacks significant contributions to the field of science and clinical practice. Additionally, concerns have been raised regarding the sample size, patient selection criteria, and the methods used to compare different biochemical tests, which do not meet the desired standards of quality.
I understand the effort and dedication you have invested in your research, and I encourage you to address these concerns in future revisions. However, based on the current state of the manuscript, I feel compelled to advise against submission to Antibiotics.
Thank you for considering my feedback, and I wish you all the best in your academic endeavors.
Warm regards,
Author Response
Thank you for your time and consideration in reviewing our manuscript. We have taken into consideration comments provided by other reviewers and revised the manuscript substantially to more objectively define bacterial infection in our cohort and provide more information regarding clinical care decision pathway. The discussion has also been expanded. We feel these data are of clinical relevance and interest as few groups have reported a comparison between procalcitonin and calprotectin in this very specialized population.
Reviewer 2 Report
Comments and Suggestions for Authors
1, Line 26-28: “This study suggests serum calprotectin is a valuable biomarker for differentiation between types of infection and estimation of disease severity in febrile infants.” This study did not provide the evaluation on disease severity for enrolled patients. The data did not support the conclusion.
2, The captions of figure 3 and figure 4 were upside down.
3, In table 3 and table 4, the comparisons between AUC, sensitivity and specificity lacked the statistical analysis.
4, Please detailed describe diagnostic methods for bacterial and viral infections in the manuscript.
5, Among enrolled 141 patients, 54 cases (38.3%) were fever of unknown source (Line 82). Can infections be ruled out for those patients? If so many cases did not get clear diagnosis, the assessment of biomarkers was not reliable.
Author Response
We thank the reviewer for their comment and agree regarding the importance of final diagnostic decision. We have added information in the methods section where we have standardized the definition of bacterial infection per accepted characteristics described in the literature, including UTI and bacteremia. We have provided the sample sizes for each and also updated Table 1 to provide the specific bacterial species identified. For blood culture, no repeats were included. Patients with blood culture were defined as having ‘bacteremia’ unless there was evidence of organ dysfunction, which is part of the definition of pediatric sepsis (PMID: 38245889, PMID: 34612847). Decision to complete urine culture was based on clinical care pathway described. In a small portion of patients, the pathway was not adhered to by clinical team. For viral illness, this classification was based on the decision of the clinical team, including NPS PCR results and the clinical picture. Unfortunately, further information was not available in our data review.
We appreciate the reviewer comment regarding Youden’s index and have added the informedness metric for both Table 3 and Table 4. Based on reviewer comment, we also classified individuals as bacterial vs viral infection using the more objective criteria provided. This did result in changes to biomarker performance, with PCT demonstrated improved performance, particularly when considering only those with confirmed viral or bacterial infection. The discussion has been updated accordingly.
Reviewer 3 Report
Comments and Suggestions for Authors
Dear authors
The basic idea is valid, however I believe there are some methodological gaps that sould be fixed.
In a work in which the usefulness of a biomarker in differentiating between bacterial infection and other types of pathologies wants to be underlined, diagnostic confirmation is fundamental.
It is necessary to review all discharge diagnoses of "bacterial infection" and confirm them with defined criteria. Those criteria should be clearly explained in the paper.
For example: there were 7 positive blood cultures but only 4 diagnoses of sepsis, authors should better specify why this discrepancy, whether the other positive blood cultures were contaminants or simply repeated positivities.
How the other 19 confirmed bacterial infections were classified? Urinary tract infections? It is essential to discern this point (e.g. describe methods of collecting urine cultures, why in some cases they were considered and in others not).
The same must be done for defined viral infections. How were they diagnosed? What type of nasopharyngeal swabs have been considered? Bacterial, viral, PCR or culture?
As regards the tests evaluation, I believe that the Youden index (or, even better, informedness) is more explanatory than the AUC. As reported in Table 3, Youden index of calprotectin and CRP are higher than PCT and WBC, but stand at just over 0.5, which is equivalent to little more than a coin toss.
The performances improve in table 4 but I believe that this needs to be recalculated in light of more objective diagnostic criteria than the mere decision of the clinician.
Author Response
- Line 26-28: “This study suggests serum calprotectin is a valuable biomarker for differentiation between types of infection and estimation of disease severity in febrile infants.” This study did not provide the evaluation on disease severity for enrolled patients. The data did not support the conclusion.
Thank you for this important comment. We understand the reviewer’s concern given that severity was not directly assessed in this study. We have updated the conclusion accordingly and also stated it as a limitation in the relevant section.
- The captions of figure 3 and figure 4 were upside down.
We have corrected this error.
- 3, In table 3 and table 4, the comparisons between AUC, sensitivity and specificity lacked the statistical analysis.
We have updated the data analysis to compare across AUC across biomarkers using DeLong’s test for correlated ROC curves (Biometrics. 1988 Sep;44(3):837-45. PMID: 3203132). Pairwise comparisons were completed between each biomarker relative to calprotectin and are discussed in the results section.
- Please detailed describe diagnostic methods for bacterial and viral infections in the manuscript.
We appreciate both reviewer’s comments regarding the diagnostic methodology in our study. We have provided detailed information regarding the clinical pathway to final diagnosis and the criteria/definitions established on. To standardize definitions to the literature, we have updated our diagnostic classifications rather than adhere to clinician decision. This includes differentiating UTI, bacteremia, and sepsis. This led to some reclassifications of final diagnoses as compared to initial analysis, resulting in a high number of classified bacterial infections. We have updated our analysis accordingly as well as the discussion and results throughout.
- Among enrolled 141 patients, 54 cases (38.3%) were fever of unknown source (Line 82). Can infections be ruled out for those patients? If so many cases did not get clear diagnosis, the assessment of biomarkers was not reliable.
In these patients, fever could not be traced to positive bacterial culture or viral PCR testing. This is not uncommon in newborns as fever may arise in many clinical scenarios. This may include self limiting viral illness that does not need any treatment and disappears without sequelae (BMJ. 2003 Nov 8;327(7423):1094-7). To address potential contamination of the diagnostic performance assessment, we also compared biomarkers while excluding this group of patients (Table 4). While this limited the sample size, it allowed for an unbiased assessment.
Round 2
Reviewer 1 Report
Comments and Suggestions for Authors
Dear Authors
Thank you for the invitation. I congratulate the authors on the changes to the manuscript. However, the flaws I mentioned in my previous comment are still present, and I regret to inform you that this article is not suitable for publication in Antibiotics.
In particular, concerns were raised about sample size, patient selection criteria and the methods used to compare different biochemical tests that did not meet the required quality standards.
On the other hand, I think it is necessary to have a new and novel marker in addition to biomarkers that are frequently used in the clinic and which have proved their importance in terms of reliability and selectivity. If a similar comparison has to be performed, I think that the sample size should be much higher than 141.
Sincerely yours,
Author Response
See attached word document

Reviewer 2 Report
Comments and Suggestions for Authors
1,Line 116-118:“Calprotectin demonstrated the highest sensitivity (78.3%) relative to procalcitonin (67.6%), CRP (64.9%) and WBC count (62.1%%) in the entire cohort.” The comparison of sensitivity requires the statistics analyis.
2, This study had only 41 patients with laboratory confirmed bacterial or viral infection (Table 4, Figure 4B, n=41). The considerable small sample size has decreased the rigor and reliability of this study. Please highlight this point in the abstract and discussion section.
3, Figure 1 showed a total of 106 cases (33+24+45+4=106). However, the entire clinical cohort had 141 cases (Table 3; Figure 4A; Line 114). What’s the reason for the inconsistence?
4, In the previous manuscript, Figure 1 displayed 19 cases of bacterial infections. But in the revised manuscript, Figure 1 showed 33 cases of bacterial infections. Please explaine the alteration in detail.
Author Response
Thank you to the reviewers for their comments. See word document attached for point-by-point response.

Reviewer 3 Report
Comments and Suggestions for Authors
Dear Authors
Thank you for responding to the issues raised above.
corrections
- line 72-73-74: if the title of the article is the "generic" usefulness of biomarkers, then the focus of the study should be the same and not solely emphasize calprotectin. If, on the other hand, the focus is calprotectin (and the fact that this study shows that there is no difference with other biomarkers in differentiating between bacterial or viral infection), then the title of the article should be changed.
- line 97: correct "dephalexin" to "cephalexin"
- lines 102-109: the fact that there are statistically significant differences in biomarker levels is a bias as these were certainly used by clinicians in deciding both admission and treatment of patients. This finding is inconsistent and should be clearly stated in the text.
- Table 2: It is not indicated whether biomarkers values are mean, median, whether numbers in parentheses are IQR or 95%CI, please correct
- lines 122-124: what about between calprotectin and CRP?
- line 190: please correct to "moderate"
- lines 196-198: actually showed fair performance but lower than other commonly used biomarkers, this should be indicated
- lines 225-227: the analysis was done on actual diagnoses and not clinical judgment, so this bias should be overcome, please correct.
- lines 325-328: I do not understand this point, it has just been shown that calprotectin is at least equal to PCT (and if so, also evaluate the differences with CRP in terms of AUC between bacterial and viral infections). This seems more like a sponsor than a real conclusion: it should be clearly stated that calprotectin showed no difference with the other commonly used biomarkers.
Best regards
Author Response

(The authors gave the same response as above.)

Round 3
Reviewer 1 Report
Comments and Suggestions for Authors
Dear Authors
I have completed the peer review process for the manuscript titled "Validation of Serum Calprotectin Relative to Other Biomarkers of Infection in Febrile Infants Presenting to the Emergency Department," submitted to Antibiotics journal. After careful evaluation, I recommend major revisions before considering its publication. Below, I outline the specific areas of concern and suggestions for improvement:
While the study addresses an important topic, concerns persist regarding its originality. The novelty of the findings should be further emphasized, and additional contextualization within the existing literature is necessary.
The acknowledgment of the limited sample size is appreciated, yet further discussion on its implications and potential biases is warranted. Additionally, a clearer elucidation of the study's limitations would enhance transparency and interpretation of the results.
The study contrasts calprotectin with established biomarkers like CRP and procalcitonin. However, the efficacy of calprotectin in viral infections, particularly in the context of COVID-19, remains contentious. Providing robust empirical data to support the arguments presented is essential to strengthen the study's validity and relevance.
Some sections of the figures are difficult to comprehend and appear contradictory, particularly regarding the comparison of procalcitonin levels with viral infection. Clarification and consistency in figure presentation, including the use of p-values (e.g., p<0.001), would enhance understanding and interpretation.
The scale for procalcitonin appears disproportionately larger compared to calprotectin and CRP in the figures. Adjusting the scale to reflect reference levels more accurately would improve visual clarity and facilitate comparison.
In clinical practice, biomarkers like calprotectin, CRP, WBC, and procalcitonin are often measured more frequently throughout a patient's hospitalization. Comparing biomarker levels at admission versus discharge alone may not capture the full extent of calprotectin's effects. Consideration of all biomarker levels obtained during hospitalization would provide a more comprehensive assessment.
The manuscript could benefit from a more explicit discussion on how calprotectin levels may influence antibiotic prescribing practices. Further elaboration on the clinical implications of calprotectin's utility in guiding treatment decisions is necessary.
The rationale for using the McNemar test for sensitivity and specificity calculations is unclear, particularly given the continuous nature of calprotectin, CRP, WBC, and procalcitonin levels. Consulting a biostatistics expert and considering alternative methods such as the Mann-Whitney U test would ensure appropriate statistical analysis.
Overall, while the manuscript presents valuable insights into the validation of serum calprotectin as a biomarker of infection in febrile infants, addressing the aforementioned concerns through major revisions is imperative to enhance its quality and suitability for publication.
Sincerely,
Author Response
Please see attached file for reviewer responses.

Reviewer 2 Report
Comments and Suggestions for Authors
Glad to see the correction of statistical and concept errors in the latest revised manuscript.
Line 127-128: "Procalcitonin and CRP demonstrated statistically significantly higher specificity relative to calprotectin". This point should be displayed in the abstract of the manuscript.
Author Response
Thank you again or your comments regarding our manuscript. We have added the recommended sentence to the manuscript abstract.
Reviewer 3 Report
Comments and Suggestions for Authors
Dear authors
Thanks for your replies
Best regards
Author Response
Thanks for valuable remarks, which have helped us to improve our manuscript.